# Recovering the Conformal Limit of Color Superconducting Quark Matter within a Confining Density Functional Approach

Oleksii Ivanytskyi * and David B. Blaschke *

Institute of Theoretical Physics, University of Wroclaw, Max Born Plac 9, 50-204 Wroclaw, Poland
* Correspondence: oleksii.ivanytskyi@uwr.edu.pl (O.I.); david.blaschke@uwr.edu.pl (D.B.B.)

**Abstract:** We generalize a recently proposed confining relativistic density-functional approach to the case of density-dependent vector and diquark couplings. The particular behavior of these couplings is motivated by the non-perturbative gluon exchange in dense quark matter and provides the conformal limit at asymptotically high densities. We demonstrate that this feature of the quark matter EoS is consistent with a significant stiffness in the density range typical for the interiors of neutron stars. In order to model these astrophysical objects, we construct a family of hybrid quark-hadron EoSs of cold stellar matter. We also confront our approach with the observational constraints on the mass–radius relation of neutron stars and their tidal deformabilities and argue in favor of a quark matter onset at masses below 1.0 $M_\odot$.

**Keywords:** quark matter; conformal limit; neutron stars





## 1. Introduction

Modern multi-messenger observations of neutron stars (NSs) and their mergers provide new measurements of their masses and radii. These data are important constraints on the equation of state (EoS) of cold, dense matter in a region of the QCD phase diagram which is inaccessible to ab initio simulations of QCD on the lattice or heavy-ion collision experiments. The results of the analysis give at a mass of 1.4 $M_\odot$ a radius $R_{1.4} = 11.7^{+0.86}_{-0.81}$ km [1] and at 2.0 $M_\odot$ a radius of $R_{2.0} = 13.7^{+2.6}_{-1.5}$ km [2]. These results imply that the neutron star matter EoS should not be too stiff at densities below twice the saturation density $n_0 = 0.15$ fm$^{-3}$ (roughly corresponding to the central density of an NS with M = 1.4 $M_\odot$), but has to be stiff enough at higher densities to allow for a maximum mass above 2.0 $M_\odot$. These new constraints on the NS mass–radius relation can be fulfilled within the purely nucleonic scenario for the NS interiors [3]. At the same time, approaches based on realistic nuclear interactions imply appearance of hyperons in the NS interiors, which softens EoS of nuclear matter and lowers the NS maximum mass $M_{max}$. For example, ab initio Brueckner–Hartree–Fock and cluster variational methods with the microscopic interaction potentials fitted to the nucleus–nucleus scattering data and properties of hypernuclei yield $M_{max}$ barely reaching 2.0 $M_\odot$ [4]. The analysis performed within a set of EoSs derived from relativistic density functional theory constrained by the results of chiral effective field theory, terrestrial experiments, astrophysical observations and reproducing hyperon potentials in the symmetric nuclear matter at saturation density provides marginal agreement with the NICER constraints on the NS maximum mass [2,5]. Stiffer hadronic EoSs provide better agreement even in the presence of hyperons, e.g., DD2 EoS with hyperons yields $M_{max} = 2.1$ $M_\odot$ [6]. However, stiff hadronic EoSs are discriminated by the requirement that tidal deformability of a 1.4 $M_\odot$ NS falls within the range $\Lambda_{1.4} = 70$–580 extracted from the analysis of GW170817 [7]. These complications are naturally removed when the scenario with a low onset density for the transition to stiff quark matter in the NS core is considered, so that all the above conditions can be fulfilled simultaneously. It is important to note that recent model agnostic statistical analyses report the viability of EoSs without

a strong first order phase transition [8,9]. In this case the sharp interface between quark and hadron matter is smoothened, e.g., by inhomogeneous pasta structures [10]. Moreover, an early onset of deconfinement for star masses around 0.5 $M_\odot$ could at present not be discriminated observationally from a sequence without a phase transition, even with the recent measurement of a strangely light neutron star [11]. This spectacular measurement is in excellent agreement with the scenario of an early onset of deconfinement.

As has been discussed in detail in the recent review by Baym et al. [12], an NS EoS with stiff quark matter requires a repulsive vector meanfield and strong color superconductivity with sufficiently large diquark pairing gap for the early onset of the deconfinement transition. When one aims at a sufficiently general formulation of the quark matter EoS which should also be suitable for the description of systems at finite temperatures like in supernova explosions or neutron star mergers, a confining relativistic density functional approach has proven successful [13]. The model developed in [13] has recently been generalized in [14] so that its Lagrangian obeys chiral symmetry and describes color superconductivity.

However, in these approaches, the vector meanfield persists at high densities and thus the quark matter EoS remains stiff with a squared sound speed well exceeding the conformal limit value of $c_s^2 = 1/3$. Many authors do not recognize this situation as a problem since the densities at which perturbative QCD (pQCD) provides a reliable EoS model are about one order of magnitude larger than the central densities in the most massive NSs. Nevertheless, it has been shown recently [15] that pQCD actually can constrain the EoS at NS densities, just by demanding thermodynamic stability and causality. Therefore, in the present work we want to present a possible generalization of the RDF approach to dense quark matter which recovers the conformal limit at high densities.

The organization of the paper is as follows. In the next section, we generalize the RDF approach developed in Ref. [14] to the case of density-dependent vector and diquark couplings. The EoS of cold, color-superconducting quark matter is modelled in Section 3. Its convergence to the conformal limit is analyzed in the same section. Section 4 is devoted to application of the developed EoS to modelling compact stars with quark cores. The results are summarized and discussed in Section 5.

## 2. Confining RDF Approach with Density-Dependent Vector and Diquark Couplings

A generalization of the RDF approach for the description of color-superconducting quark matter to the case of density-dependent vector and diquark couplings can be performed within the Lagrangian formalism developed in Refs. [14,16]. In the case of two quark flavors, the fundamental dynamical variables of the approach are quark fields represented by the flavor spinor $q^T = (u, d)$. We note that a first application of the present approach to the three-flavor case was given in Ref. [17]. The interaction terms are chosen in the contact current–current form $(\bar{q}\hat{\Gamma}q)^2$ with $\hat{\Gamma}$ being an interaction vertex. In the case of scalar ($\hat{\Gamma} = 1$) and pseudoscalar ($\hat{\Gamma} = i\gamma_5\vec{\tau}$) channels the corresponding quark bilinears are composed into the chirally symmetric combination $(\bar{q}q)^2 + (\bar{q}i\gamma_5\vec{\tau}q)^2$, providing the corresponding symmetry of interaction. The model Lagrangian can be written as

$$\mathcal{L} = \bar{q}(i\slashed{\partial} - m)q - \mathcal{U} + \mathcal{L}_V + \mathcal{L}_D, \tag{1}$$

where $m$ is the current mass of the two light quark flavors. The potential $\mathcal{U}$ accounts for the attractive chirally symmetric interaction in scalar and pseudoscalar channels

$$\mathcal{U} = D_0 \left[ (1 + \alpha)\langle\bar{q}q\rangle_0^2 - (\bar{q}q)^2 - (\bar{q}i\gamma_5\vec{\tau}q)^2 \right]^{\frac{1}{3}}, \tag{2}$$

where constants $D_0$ and $\alpha$ control the interaction strength and constituent quark mass in the vacuum [14,16], respectively, while $\langle\bar{q}q\rangle_0$ is the vacuum value of the chiral condensate. In what follows the subscript index "0" denotes the quantities defined in the vacuum. The present parameterization of $\mathcal{U}$ is motivated by the string-flip model (SFM) [18,19], which assumes that the interparticle interaction energy is proportional to mean separation

between quarks. The model Lagrangian includes terms representing vector repulsion and diquark pairing interactions

$$
\begin{aligned}
\mathcal{L}_V &= -G_V(\bar{q}\gamma_\mu q)^2 + \Theta_V, \\
\mathcal{L}_D &= G_D(\bar{q}i\gamma_5\tau_2\lambda_A q^c)(\bar{q}^c i\gamma_5\tau_2\lambda_A q) - \Theta_D,
\end{aligned}
$$
(3)
(4)

where charge conjugated quark field is $q^c = i\gamma_2\gamma_0\bar{q}^T$ and $A = 2, 5, 7$ labels the antisymmetric generators $\lambda_A/2$ of the SU(3) color group, so that the ansatz (4) for the diquark current fulfills the Pauli principle for the quark pair. These interaction channels are important for compact star phenomenology [12]. In Refs. [14,16,20] the couplings $G_V$ and $G_D$ were set to be constants. Here, we consider them as medium dependent functions. This section presents a general treatment, while the specific parameterization of the vector and diquark couplings adopted in this work is considered in Section 3. The naive introduction of a medium dependence for $G_V$ and $G_D$ can break thermodynamic consistency by violating thermodynamic identities similarly to the case of the naive introduction of a medium dependent dispersion relation [21]. In order to circumvent this problem we follow the strategy of Ref. [22] and introduce the so-called rearrangement terms $\Theta_V$ and $\Theta_D$ into Equations (3) and (4). Similar to $G_V$ and $G_D$, they are some medium dependent functions which should be defined in agreement with the corresponding couplings. These rearrangement terms vanish at constant vector and diquark couplings. Their signs in Equations (3) and (4) are conventional. The present choice is motivated by the fact that the corresponding terms in the Lagrangian represent repulsive and attractive interactions.

Expanding the potential $\mathcal{U}$ around the mean-field expectation values $\langle\bar{q}q\rangle \neq 0$ and $\langle\bar{q}i\gamma_5\vec{\tau}q\rangle = 0$ up to the second order terms and inserting the result to Equation (1) yields an effective Lagrangian. At this order, the only non-vanishing expansion coefficients are

$$
\Sigma_{MF} = \frac{\partial\mathcal{U}_{MF}}{\partial\langle\bar{q}q\rangle},
$$
(5)

$$
G_S = -\frac{1}{2}\frac{\partial^2\mathcal{U}_{MF}}{\partial\langle\bar{q}q\rangle^2},
$$
(6)

$$
G_{PS} = -\frac{1}{6}\frac{\partial^2\mathcal{U}_{MF}}{\partial\langle\bar{q}i\gamma_5\vec{\tau}q\rangle^2}.
$$
(7)

Hereafter, the subscript index "$MF$" denotes the quantities defined by the mean field approximation. The resulting effective Lagrangian has the current–current interaction form of the NJL type models,

$$
\begin{aligned}
\mathcal{L}_{\text{eff}} &= \bar{q}(i\slashed{\partial} - m^*)q + G_S(\bar{q}q - \langle\bar{q}q\rangle)^2 + G_{PS}(\bar{q}i\gamma_5\vec{\tau}q)^2, \\
&+ \mathcal{L}_V + \mathcal{L}_D - \mathcal{U}_{MF} + \langle\bar{q}q\rangle\Sigma_{MF},
\end{aligned}
$$
(8)

where $m^* = m + \Sigma_{MF}$ is the constituent quark mass. It follows from this effective Lagrangian that $\Sigma_{MF}$ is nothing else than scalar self-energy of the quarks at the mean-field level, while $G_S$ and $G_{PS}$ correspond to the effective couplings of quark interaction in the scalar and pseudoscalar channels, respectively. These couplings do not coincide in the general case. This corresponds to an explicit violation of chiral symmetry which results from expanding the Lagrangian $\mathcal{L}$ around the mean-field solution, which is known to be chirally broken. At the same time, the dynamical restoration of chiral symmetry at high temperatures and densities leads to the asymptotic coincidence of $G_S$ and $G_{PS}$ [14,16]. With the effective Lagrangian $\mathcal{L}_{\text{eff}}$, the partition function can be represented as a functional integral over quark fields

$$
\mathcal{Z} = \int \mathcal{D}\bar{q}\,\mathcal{D}q\,\exp\left[\int dx_E(\mathcal{L}_{\text{eff}} + q^+\hat{\mu}q)\right],
$$
(9)

where integration over the Euclidean space-time is limited to the inverse temperature $1/T \equiv \beta = \int d\tau$ and the volume $V = \int d\mathbf{x}$. The diagonal matrix $\hat{\mu} = \mathrm{diag}(\mu_u, \mu_d)$ stands for the quark chemical potentials. They can be expressed through the baryonic $\mu_B$ and electric $\mu_Q$ chemical potentials as $\mu_f = \mu_B/3 + Q_f \mu_Q$, where subscript index $f = u, d$ labels quark flavors and $Q_f$ is their electric charge.

The next step corresponds to bosonizing the partition function by means of the Hubbard–Stratonovich transformation. This introduces collective scalar ($\sigma$), pseudoscalar ($\vec{\pi}$), vector ($\omega_\mu$) and complex scalar diquark ($\Delta_A$) fields. They are coupled to the corresponding bilinears of quark fields, $\bar{q}q - \langle \bar{q}q \rangle$, $\bar{q}i\gamma_5\vec{\tau}q$, $\bar{q}\gamma_\mu q$ and $\bar{q}i\gamma_5\tau_2\lambda_A q$, respectively. It is worth noticing that the medium dependence of the couplings $G_S$, $G_{PS}$, $G_V$ and $G_D$ does not affect this procedure since none of them includes any dynamical variable. It is convenient to treat the bosonized partition function within Nambu–Gorkov formalism. Here, we just outline the main aspects of the formalism and summarize the results. The interested readers are referred to Refs. [14,23]. In this case, quark fields are collected to the Nambu–Gorkov bispinor $\mathcal{Q}^T = (q\, q^c)/\sqrt{2}$, while the partition function becomes

$$
\begin{aligned}
\mathcal{Z} \;=\; & \exp\left[\beta V\left(-\mathcal{U}_{MF} + \langle \bar{q}q \rangle \Sigma_{MF} + \Theta_V - \Theta_D\right)\right] \int \mathcal{D}\overline{\mathcal{Q}}\,\mathcal{D}\mathcal{Q}\,\mathcal{D}\sigma\,\mathcal{D}\vec{\pi}\,\mathcal{D}\omega_\mu\,\mathcal{D}\Delta_A\,\mathcal{D}\Delta_A^* \\
& \times\; \exp\left[\int dx_E\left(\overline{\mathcal{Q}}\,\mathcal{S}^{-1}\mathcal{Q} + \sigma\langle \bar{q}q \rangle - \frac{\sigma^2}{4G_S} - \frac{\vec{\pi}^2}{4G_{PS}} + \frac{\omega_\mu \omega^\mu}{4G_V} - \frac{\Delta_A^* \Delta_A}{4G_D}\right)\right].
\end{aligned}
\tag{10}
$$

Here, the propagator of the Nambu–Gorkov bispinors reads

$$
\mathcal{S}^{-1} = \begin{pmatrix} S_+^{-1} - \sigma - i\gamma_5\vec{\tau}\cdot\vec{\pi} & i\Delta_A\gamma_5\tau_2\lambda_A \\ i\Delta_A^*\gamma_5\tau_2\lambda_A & S_-^{-1} - \sigma - i\gamma_5\vec{\tau}^T\cdot\vec{\pi} \end{pmatrix},
\tag{11}
$$

with $S_\pm^{-1} = i\slashed{\partial} \pm \slashed{\varphi} - m^* \pm \gamma_0\hat{\mu}$. The exponential in the second line of Equation (10) is nothing else than the quark-meson part of the bosonized action. The quark fields enter this action quadratically and can therefore be integrated out analytically yielding $\mathrm{Tr}\ln\left(\beta\mathcal{S}^{-1}\right)/2$ in the exponential. The trace $\mathrm{Tr}$ hereafter is performed over the color, flavor, Dirac, three-momentum and Matsubara indices. The last ones appear after going over to the momentum representation, which yields $S_\pm^{-1} = \slashed{k} - m^*$ with $k_0 = iz_n \pm \hat{\mu}^*$, $z_n = (2n+1)\pi T$, defining a fermionic Matsubara frequency and $\mu_f^* = \mu_f + \omega$ being effective chemical potential of quarks.

The action in Equation (10) gives direct access to the Euler–Lagrange equations of the scalar, pseudoscalar, vector and diquark fields. Averaging these equations for the vector and diquark fields, one obtains $\langle \omega_\mu \rangle = -2G_V\langle \bar{q}\gamma_\mu q \rangle$ and $\langle \Delta_A \rangle = 2G_D\langle \bar{q}^c i\tau_2\gamma_5\lambda_A q \rangle$, respectively. By a proper Lorentz transform, the vector field average attains the form $\langle \omega_\mu \rangle = g_{\mu 0}\omega$ with $\omega = -2G_V\langle q^+ q \rangle$. Furthermore, there exists a global color rotation which leaves $\Delta_2$ as the only diquark field with a nonvanishing expectation value at the mean field level. We note that only its modulus $\Delta = |\Delta_2|$ appears in the expression for thermodynamic potential. Averaging the Euler–Lagrange equations for scalar and pseudoscalar fields shows that $\langle \sigma \rangle$ and $\langle \vec{\pi} \rangle$ vanish at mean field [14]. Thus, $\sigma$ and $\vec{\pi}$ have beyond mean-field nature and represent the corresponding mesonic correlations of quarks. Within the Gaussian approximation, the back-reaction of these correlations on the quark propagator is neglected [23]. This allows to expand $\mathrm{Tr}\ln\left(\beta\mathcal{S}^{-1}\right)$ up to the second order in $\sigma$ and $\vec{\pi}$. The second order terms are quadratic in the mean-field quark propagator $\mathcal{S}_{MF}$ and thus represent one-loop polarization operators of (pseudo)scalar mesons. The latter can be used in order to construct mesonic propagators and to extract the corresponding masses from the position of the propagator poles. Within the generalized Beth–Uhlenbeck approach, mesonic propagators also can be used in order to obtain beyond mean-field contributions to the thermodynamic potential [16,23]. In the present work, however, they are neglected because we restrict ourselves to the mean-field approximation. For this, we

replace scalar, pseudoscalar, vector and diquark fields in Equation (10) by their expectation values and reduce the corresponding functional integrals. This yields

$$\Omega = -\frac{\ln \mathcal{Z}}{2\beta V} = \Omega_q + \mathcal{U}_{MF} - \langle \bar{q}q \rangle \Sigma_{MF} - \frac{\omega^2}{4G_V} + \frac{\Delta^2}{4G_D} - \Theta_V + \Theta_D. \tag{12}$$

The first term in this expression is due to the contribution of quark quasiparticles

$$\Omega_q = -\frac{T}{2V} \text{Tr} \ln(\beta \mathcal{S}_{MF}^{-1}) = -2 \sum_{f,c,a=\pm} \int \frac{d\mathbf{k}}{(2\pi)^3} \left[ \frac{g_\mathbf{k}}{2} \epsilon_{\mathbf{k}fc}^a - T \ln\left(1 - f_{\mathbf{k}fc}^a\right) \right]. \tag{13}$$

It includes the single particle energies shifted by the effective chemical potential and distribution functions, i.e.,

$$\epsilon_{\mathbf{k}fc}^{\pm} = \text{sgn}(\epsilon_{\mathbf{k}f} \mp \mu_f^*) \sqrt{(\epsilon_{\mathbf{k}f} \mp \mu_f^*)^2 + \Delta_c^2} \quad \text{and} \quad f_{\mathbf{k}fc}^{\pm} = \left[ e^{\beta \epsilon_{\mathbf{k}fc}^{\pm}} + 1 \right]^{-1}. \tag{14}$$

Here, $\epsilon_{\mathbf{k}f} = \sqrt{\mathbf{k}^2 + m^{*2}}$ and the subscript index $c = r, g, b$ labels quark color states. The color vector $\Delta_c = (\Delta, \Delta, 0)$ is introduced in order to unify the notations and $a = \pm$ distinguishes particles and antiparticles. The dispersion relation (14) can be obtained by solving $\det(\mathcal{S}_{MF}^{-1}) = 0$ with respect to the zeroth component of quark four momentum $k$. It shows that only red and green quarks are paired exhibiting the gap $\Delta$ in their one-particle energy spectrum, while blue quarks are unpaired. The zero point terms in the expression for $\Omega_q$ are regularized by smooth cut-off in the Gaussian form

$$g_\mathbf{k} = \exp\left[ -\frac{\mathbf{k}^2}{\Lambda^2} \right]. \tag{15}$$

In Ref. [14], such a form was chosen in order to prevent a discontinuous behavior of various thermodynamic quantities which would have been obtained for the 2SC phase of quark matter with a sharp cutoff as $g_\mathbf{k} = \theta(\Lambda - |\mathbf{k}|)$.

The thermodynamic definition of the number density of a given quark flavor corresponds to the thermodynamic identity $\langle f^+ f \rangle = -\partial \Omega / \partial \mu_f$, which should be used carefully since vector and diquark couplings are medium dependent functions. On the other hand, the statistical definition of this quantity implies $\langle f^+ f \rangle = -\partial \Omega_q / \partial \mu_f$. The thermodynamic consistency of the present approach is provided when these two definitions coincide. Thus, we require

$$\frac{\partial \Omega}{\partial \mu_f} - \frac{\partial \Omega_q}{\partial \mu_f} = \frac{\omega^2}{4G_V^2} \frac{\partial G_V}{\partial \mu_f} - \frac{\Delta^2}{4G_D^2} \frac{\partial G_D}{\partial \mu_f} - \frac{\partial \Theta_V}{\partial \mu_f} + \frac{\partial \Theta_D}{\partial \mu_f}$$

$$= \langle q^+ q \rangle^2 \frac{\partial G_V}{\partial \mu_f} - \frac{\partial \Theta_V}{\partial \mu_f} - |\langle \bar{q}^c i\tau_2 \gamma_5 \lambda_2 q \rangle|^2 \frac{\partial G_D}{\partial \mu_f} + \frac{\partial \Theta_D}{\partial \mu_f} = 0, \tag{16}$$

where the mean-field equations $\omega = -2G_V \langle q^+ q \rangle$ and $\Delta = 2G_D |\langle \bar{q}^c i\tau_2 \gamma_5 \lambda_2 q \rangle|$ were used on the second step. Fulfilment of this condition requires the rearrangement terms $\Theta_V$ and $\Theta_D$ to be defined in accordance with the couplings $G_V$ and $G_D$. The corresponding relations can be easily found by assuming that $\Theta_V$, $G_V$, $\Theta_D$ and $G_D$ are functions of $\langle q^+ q \rangle$ and $|\langle \bar{q}^c i\tau_2 \gamma_5 \lambda_2 q \rangle|$, respectively. In this case, Equation (16) leads to

$$\Theta_V = \int\limits_0^{\langle q^+ q \rangle} dn \, n^2 \frac{\partial G_V(n)}{\partial n} \quad \text{and} \quad \Theta_D = \int\limits_0^{|\langle \bar{q}^c i\tau_2 \gamma_5 \lambda_2 q \rangle|} dn \, n^2 \frac{\partial G_D(n)}{\partial n}. \tag{17}$$

From these relations, it is seen that the rearrangement terms, indeed, vanish if the couplings are constant. Using these relations number density of a given quark flavor, chiral

condensate and modulus of the diquark, one can be found from the quark part of the thermodynamic potential as

$$\langle f^+ f \rangle = -\frac{\partial \Omega_q}{\partial \mu_f} = 2 \sum_{c,a=\pm} a \int \frac{d\mathbf{k}}{(2\pi)^3} \left( f^a_{\mathbf{k}fc} - \frac{g_{\mathbf{k}}}{2} \right) \left( 2\Delta_c \delta(\epsilon^a_{\mathbf{k}fb}) + \frac{\epsilon^a_{\mathbf{k}fb}}{\epsilon^a_{\mathbf{k}fc}} \right), \quad (18)$$

$$\langle \bar{q}q \rangle = \frac{\partial \Omega_q}{\partial m} = 2 \sum_{f,c,a=\pm} \int \frac{d\mathbf{k}}{(2\pi)^3} \left( f^a_{\mathbf{k}fc} - \frac{g_{\mathbf{k}}}{2} \right) \left( 2\Delta_c \delta(\epsilon^a_{\mathbf{k}fb}) + \frac{\epsilon^a_{\mathbf{k}fb}}{\epsilon^a_{\mathbf{k}fc}} \right) \frac{m^*}{\epsilon_{\mathbf{k}f}}, \quad (19)$$

$$|\langle \bar{q}^c i\tau_2 \gamma_5 \lambda_2 q \rangle| = -\frac{\partial \Omega_q}{\partial \Delta} = 2 \sum_{f,c,a=\pm} \int \frac{d\mathbf{k}}{(2\pi)^3} \left( \frac{g_{\mathbf{k}}}{2} - f^a_{\mathbf{k}fc} \right) \frac{\Delta_c}{\epsilon^a_{\mathbf{k}fc}}. \quad (20)$$

We note that the Dirac delta-function in Equations (18) and (19) appears due to differentiating the sign-function from the dispersion relation (14). It is also worth mentioning that the definitions of the rearrangement terms given by Equation (17) along with the mean-field equations $\omega = -2G_V \langle q^+ q \rangle$ and $\Delta = 2G_D |\langle \bar{q}^c i\tau_2 \gamma_5 \lambda_2 q \rangle|$ are sufficient in order to obtain the number density of a given quark flavor, the chiral condensate and the modulus of the diquark one in the form (18)–(20). This holds for any functional dependence of the couplings $G_V$ and $G_D$ on their arguments. With Equations (18) and (20), the mean-field equations for the vector field and diquark pairing gap can be given an explicit form. Furthermore, Equation (19) should be understood as another mean-field equation with respect to chiral condensate. Its solution along with the solutions of the mean-field equations for vector field and pairing gap minimize the thermodynamic potential. For the reader's convenience, in Appendix A we explicitly analyze the conditions providing the minimum of $\Omega$ and derive from them the mean-field equations mentioned above as well as Equations (18)–(20).

Once these mean-field equations are consistently solved, pressure, entropy and energy density can be found using the thermodynamic identities $p = \Omega_0 - \Omega$, $s = \partial p / \partial T$ and $\varepsilon = \sum_f \mu_f \langle f^+ f \rangle + Ts - p$, while squared speed of sound is defined as the derivative $c_S^2 = dp/d\varepsilon$ calculated at constant entropy. Below we also analyze the dimensionless interaction measure $\delta = 1/3 - p/\varepsilon$ being nothing other than the trace of the energy momentum tensor scaled by the conformal limit for the pressure which is $3\varepsilon$.

It is worth noticing that the present approach with density-dependent vector and diquark couplings is equivalent to a density functional approach in the spirit of Ref. [13]. In the case of vector repulsion and diquark pairing, the corresponding density functionals depend on the quark bilinears $q^+ q$ and $\bar{q}^c i\tau_2 \gamma_5 \lambda_2 q$, $\bar{q} i\tau_2 \gamma_5 \lambda_2 q^c$, respectively. Consistency with the present approach with medium dependent vector and diquark couplings is provided by

$$\mathcal{U}_V = \int\limits_0^{(q^+ q)^2} dn^2 \, G_V(n) \quad \text{and} \quad \mathcal{U}_D = \int\limits_0^{|\bar{q}^c i\tau_2 \gamma_5 \lambda_2 q|^2} dn^2 \, G_D(n). \quad (21)$$

Expanding these potentials around the mean-field solutions up to the first order terms produces the vector and diquark self-energies of quarks

$$\Sigma_V \equiv \frac{\partial \mathcal{U}_{V,MF}}{\partial \langle q^+ q \rangle} \quad \text{and} \quad \hat{\Sigma}_D \equiv \text{antidiag} \left( \frac{\partial \mathcal{U}_{D,MF}}{\partial \langle \bar{q} i\tau_2 \gamma_5 \lambda_2 q^c \rangle}, \frac{\partial \mathcal{U}_{D,MF}}{\partial \langle \bar{q}^c i\tau_2 \gamma_5 \lambda_2 q \rangle} \right), \quad (22)$$

where $\hat{\Sigma}_D$ is an antidiagonal matrix in the Nambu–Gorkov space due to the fact that $\mathcal{U}_D$ depends on two dynamical variables $\bar{q}^c i\tau_2 \gamma_5 \lambda_2 q$ and $\bar{q} i\tau_2 \gamma_5 \lambda_2 q^c$. The vector self-energy shifts the quark chemical potential by exactly the same amount as $\omega = -2G_V \langle q^+ q \rangle$. The corresponding pressure term $-\mathcal{U}_{V,MF} + \langle q^+ q \rangle \Sigma_V$ also coincides with $\omega^2 / 4G_V + \Theta_V$. The diquark self-energy coincides with the non-diagonal terms in the inverse Nambu–Gorkov propagator if the diquark fields in Equation (11) are replaced by their expectation values discussed above. In this case, the pressure term coming from the expansion of the diquark potential $-\mathcal{U}_D + \langle \bar{\mathcal{Q}} i\tau_2 \gamma_5 \lambda_2 \hat{\Sigma}_D \mathcal{Q} \rangle$ coincides with $-\Delta^2/(4G_D) - \Theta_D$.

The present model has four parameters relevant to the QCD phenomenology, which are $m$, $D_0$, $\alpha$ and $\Lambda$ [14]. The pion mass $M_\pi$ and decay constant $F_\pi$ are the most important observables in this context. An analysis of the scalar mode mass $M_\sigma$ also was performed despite the fact that its experimental status is far from being clear. Our approach allows $M_\sigma$ in a wide interval covering the masses of all the experimental candidates. Typically, the lightest state $f_0(500)$ is considered as a candidate for the scalar meson role. It, however, has a large decay width of about 500–1000 MeV [24] and should be considered rather a tetraquark state than a traditional quark–antiquark meson [25]. Therefore, it is not appropriate to fit the vacuum parameters of the low-energy QCD model using the $f_0(500)$ state as a quark–antiquark meson. Our analysis uses instead the $f_0(980)$ state as a scalar meson. Our approach does not fit the vacuum value of chiral condensate per flavor $|\langle \bar{l}l \rangle_0^{1\,GeV}|^{1/3} = 241$ MeV found from QCD sum rules at the renormalization scale 1 GeV [26]. This problem is typical for most of the chiral quark matter models [27]. Therefore, within the present approach we allowed the chiral condensate to have a somewhat larger value in order to have a reasonable value of the pseudocritical temperature $T_{PC} = 163$ MeV defined by the peak position of chiral susceptibility. The model parameters defined using the above strategy along with the resulting physical quantities are presented in Table 1. This parameter set yields $m^* = 718$ MeV in the vacuum, which provides an efficient phenomenological confinement of quarks due to their high masses at low temperatures and densities.

**Table 1.** Parameters of the present model and resulting observables.

| $m$ (MeV) | $\Lambda$ (MeV) | $\alpha$ | $D_0\Lambda^{-2}$ | $M_\pi$ (MeV) | $F_\pi$ (MeV) | $M_\sigma$ (MeV) | $|\langle \bar{l}l \rangle_0|^{1/3}$ (MeV) |
|---|---|---|---|---|---|---|---|
| 4.2 | 573 | 1.43 | 1.39 | 140 | 90 | 980 | 267 |

The parameterization of the vector coupling adopted in this work is motivated by the analysis of the quark repulsion energy due to non-perturbative gluon exchange of QCD in the Landau gauge [28]. In the normal phase of symmetric quark matter, this implies $G_V \propto (9M_g^2 + 8k_F^2)^{-1}$, with $M_g$ and $k_F = (\pi^2\langle q^+q \rangle/2)^{1/3}$ being the non-perturbative gluon mass and the quark Fermi momentum, respectively. With this, we introduce

$$G_V = \frac{G_{V0}}{1 + \frac{8}{9M_g^2}\left(\frac{\pi^2\langle q^+q \rangle}{2}\right)^{2/3}}, \tag{23}$$

$$G_D = \frac{G_{D0}}{1 + \frac{8}{9M_g^2}\left(\frac{\pi^2|\langle \bar{q}^c i\tau_2\gamma_5\lambda_2 q \rangle|}{2}\right)^{2/3}}. \tag{24}$$

At $M_g \to \infty$, this parameterization corresponds to constant vector and diquark couplings. At the same time, the solution of the gluon Schwinger–Dyson equations in the Landau gauge implies $M_g = 300$–700 MeV [29,30]. The value of $M_g$ can also be estimated based on the Shifman, Vainshtein and Zakharov expansion of the two-point current correlation functions within massive gauge invariant QCD [31]. For the frozen QCD structure constant $\alpha_s = 0.2$ and transferred momentum $Q^2 = 10$ GeV$^2$ that approach yields $M_g = 750$ MeV. At $\alpha_s = \pi$, which is expected for the non-perturbative regime [32], the effective gluon mass becomes 516 MeV. At the same time, for the transferred momentum coinciding with the ultraviolet cut-off $\Lambda$ from Table 1, i.e., for $Q^2 = \Lambda^2$, one obtains $M_g = 942$ MeV at $\alpha_s = 0.2$ and $M_g = 792$ MeV at $\alpha_s = \pi$. Below, we consider several values of the non-perturbative gluon mass, covering a range that contains the values mentioned above. This allows us to demonstrate continuous convergence to the conformal limit at all finite $M_g$.

We treat the vacuum values of the vector $G_{V0}$ and diquark $G_{D0}$ couplings as free parameters. They are parameterized as dimensionless ratios defined with the vacuum value of the scalar coupling $G_{S0} = 18.1$ GeV$^{-2}$ through $\eta_V \equiv G_{V0}/G_{S0}$ and $\eta_D \equiv G_{D0}/G_{S0}$. In what follows, pairs of numbers $(\eta_V, \eta_D)$ are used in order to label different EoS parmetrizations

obtained within the present model. It is necessary to stress that the physical values of $\eta_D$ are limited from above by the value $\eta_D^{max} = (3/2)(G_{PS0}/G_{S0})m_0^*/(m_0^* - m)$, beyond which the vacuum state would already become color superconducting [14], see also [33,34]. For the chosen values of the model parameters $\eta_D^{max} = 0.78$. In order to not go to the marginal values of the diquark coupling, we constrain the consideration to the range $\eta_D < 0.77$.

### 3. Cold Quark Matter

Studying cold quark matter at vanishing temperatures is of practical interest for modeling compact stars. At $T = 0$, the single particle distribution functions of quarks reduce to unit step-functions, i.e., $f_{\mathbf{k}fc}^+ = \theta(-\epsilon_{\mathbf{k}fc}^+)$ and $f_{\mathbf{k}fc}^- = 0$. Therefore, the antiquark terms are absent at $T = 0$, while for the quark ones the integration over $\mathbf{k}$ is limited by the Fermi momentum defined by the condition $\epsilon_{\mathbf{k}fb}^+ = 0$.

Constructing the EoS of quark matter requires solving mean-field equations for chiral condensate, vector field and diquark pairing. Solutions of these equations give direct access to the single quark energies $\epsilon_{\mathbf{k}fc}^+$, which are needed in order to calculate the thermodynamic quantities mentioned in the previous section. Before considering these quantities in detail, we would like to analyze the high density asymptotics of our approach and show that it is consistent with the conformal limit of strongly interacting matter. At high densities, the zero point terms, the quark masses as well as all terms related to $\mathcal{U}$ can be neglected. In order to analyze the behavior of the pairing gap, we notice that in the considered regime $G_D \propto |\langle \bar{q}^c i\tau_2\gamma_5\lambda_2 q\rangle|^{-2/3}$. Therefore, using the pairing gap equation $\Delta = 2G_D|\langle \bar{q}^c i\tau_2\gamma_5\lambda_2 q\rangle|$ and Equation (20), we obtain

$$\Delta \simeq 2G_{D0}\frac{9M_g^2}{8}\left(\frac{2}{\pi^2}\right)^{2/3}|\langle \bar{q}^c i\tau_2\gamma_5\lambda_2 q\rangle|^{1/3} = \frac{9G_{D0}M_g^2}{(2\pi)^{4/3}}\left[-4\Delta\sum_f\int\frac{d\mathbf{k}}{(2\pi)^3}\frac{f_{\mathbf{k}fr}^+}{\epsilon_{\mathbf{k}fr}^+}\right]^{1/3}, \quad (25)$$

where the summation over the color index was performed explicitly in the second step. For $\Delta \gg \mu_f^*$, the single particle energy of paired quarks can be approximated as $\epsilon_{\mathbf{k}fr}^+ \simeq -\Delta$ and the bracket on the right hand side of this relation behaves as $\sim \sum_f \mu_f^{*3}$. This leads to $\Delta \sim (\sum_f \mu_f^{*3})^{1/3}$, which contradicts the original assumption. At $\Delta \sim \mu_f^*$ or $\Delta \ll \mu_f^*$, the bracket in Equation (25) $\sim \Delta \sum_f \mu_f^{*2}$. Thus, the high density asymptotic of the pairing gap is $\Delta \sim (\sum_f \mu_f^{*2})^{1/2}$. The corresponding contribution to the pressure behaves as $p_D \equiv -\Delta^2/4G_D - \Theta_D \sim \Delta^4$. In order to show that $p_D$ scales as the vector field term $p_V \equiv \omega^2/4G_V + \Theta_V \sim \langle q^+q\rangle^{4/3}$, we consider the quark number density

$$\langle q^+q\rangle = 2\sum_{f,c}\int\frac{d\mathbf{k}}{(2\pi)^3}f_{\mathbf{k}fc}^+\left(2\Delta_c\delta(\epsilon_{\mathbf{k}fb}^+) + \frac{\epsilon_{\mathbf{k}fb}^+}{\epsilon_{\mathbf{k}fc}^+}\right). \quad (26)$$

The first term in the bracket of this expression contributes $\sim \sum_f \Delta\mu_f^{*2}$ to the quark number density. Following the above analysis of the pairing gap, this contribution is of the same order as the second term in the brackets in Equation (26), i.e., $\sim \sum_f \mu_f^{*3} \sim \Delta^3$. With this, we conclude that $p_D \sim \langle q^+q\rangle^{4/3}$. Similarly, for the zero temperature quark pressure at high densities, we obtain

$$p_q \simeq -\Omega_q \simeq -2\sum_{f,c}\int\frac{d\mathbf{k}}{(2\pi)^3}f_{\mathbf{k}fc}^+\epsilon_{\mathbf{k}fc}^+ \sim \sum_f \mu_f^{*4} \sim \langle q^+q\rangle^{4/3}. \quad (27)$$

Thus, the total pressure at high densities is proportional to $n_B^{4/3}$ with $n_B = \langle q^+q\rangle/3$ being the baryon charge density. Along with the thermodynamic identity $n_B = \partial p/\partial\mu_B$, this leads to the high density scaling $p \sim \mu_B^4$, which respects the conformal limit. Convergence to this limit is shown in Figure 1. At small values of the baryonic chemical potentials, chiral symmetry is broken, quark masses are high and the paring gap vanishes. The pressure also

vanishes in this regime, leading to a vanishing speed of sound and an interaction measure equal to one third. At a certain $\mu_B$, the quark mass discontinuously drops to some small value, while the pairing gap becomes finite. This discontinuous change signals a first order transition from chirally broken quark matter to the 2SC phase. At this transition, the speed of sound jumps to some finite value, while the interaction measure exhibits a kink. In the general case, $\delta$ experiences a jump of the amplitude $p_c(1/\varepsilon_1 - 1/\varepsilon_2)$, where $p_c$ is pressure at the phase transition and $\varepsilon_1$ and $\varepsilon_2$ are energy densities of the coexisting low and high density phases. At zero temperature, this jump of the interaction measure degenerates to kink with $\delta = 1/3$ since $p_c = 0$ in this case. Above the transition $c_S^2$ decreases and $\delta$ has a minimum with some negative value. Position of this minimum can be found by requiring $d\delta/d\varepsilon = p/\varepsilon^2 - c_S^2/\varepsilon = 0$ or, equivalently, $c_S^2 = p/\varepsilon$. This condition can be fulfilled only if $p > 0$ just because $c_S^2$ are $\varepsilon$ and are positively defined. This explains why the minimum of $\delta$ is located in the region with positive pressure, i.e., above the transition from chirally broken quark matter to the 2SC phase. Qualitatively identical behavior of $c_S^2$ and $\delta$ in the density range with positive pressure is exhibited by the parametric model of the quark matter EoS from Ref. [35]. At any finite value of the non-perturbative gluon mass and for $\mu_B \rightarrow \infty$, the squared speed of sound and the interaction measure converge to their values in the conformal limit, $c_S^2 = 1/3$ and $\delta = 0$, respectively. It is important to stress that vanishing of the interaction measure at finite $\mu_B$ does not signal reaching the conformal limit, which also requires $\partial \delta/\partial \mu_B = 0$. Note, these two conditions necessarily lead to $c_S^2 = 1/3$ not being the case of finite values of the baryonic chemical potential. The smaller the $M_g$, the faster this convergence is. In other words, the non-perturbative gluon mass defines the scale at which the quark–quark interaction effects cease. It is also worth mentioning that within the considered model $c_S^2 \rightarrow 1/3$ from above and $\delta \rightarrow 0$ from below. This means that vector repulsion between quarks dominates the attractive pairing interaction even at high densities. An alternative scenario with $c_S^2$ approaching the conformal limit from below and $\delta$ approaching it from above requires domination of the attractive (pairing) interactions at high $\mu_B$. Such a case can be provided, e.g., if the common gluon mass parameter $M_g$ in Equations (23) and (24) is replaced by two independent vector ($M_{gV}$) and diquark ($M_{gD}$) masses with $M_{gV} < M_{gD}$. In this case, the repulsive vector interaction ceases before the attractive pairing one. An analysis of this scenario is beyond the scope of the present work. It is also important to note that for finite $M_g$ the variation of $c_S^2$ in the density region typical for NSs is much more pronounced than for $M_g \rightarrow \infty$, which corresponds to the case of constant vector and diquark couplings considered in Ref. [14]. As is seen from Figure 1, in symmetric quark matter this variation within the interval of $\mu_B$ from 1 GeV to 2 GeV is about 10%.

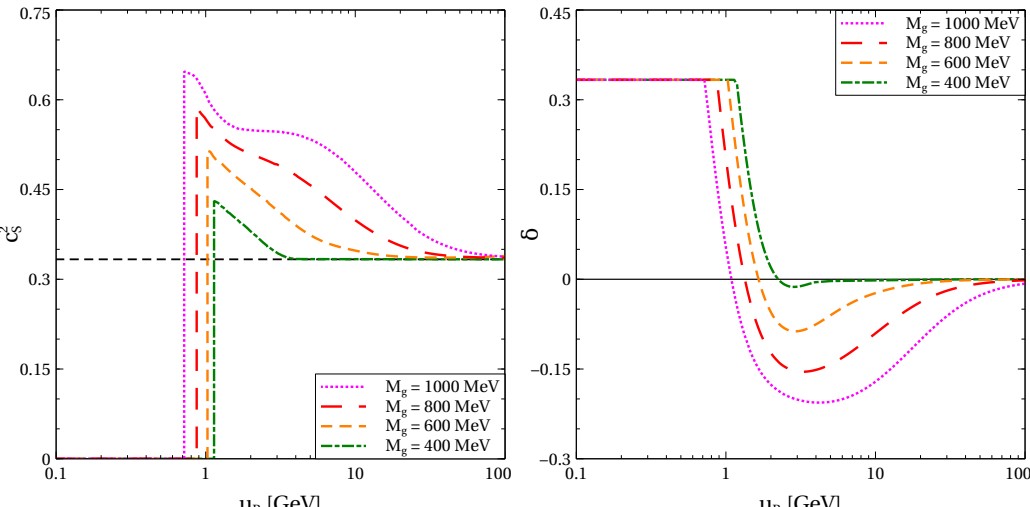

**Figure 1.** Squared speed of sound $c_S^2$ (**left panel**) and interaction measure $\delta$ (**right panel**) as functions of baryonic chemical potential $\mu_B$. The black dashed line on the left panel represents $c_S^2 = 1/3$. Calculations are performed for cold symmetric quark matter at $\eta_V = 0.32$ and $\eta_D = 0.71$.

Qualitatively the same behavior of $c_S^2$ and $\delta$ is observed at any values of $\eta_V$ and $\eta_D$ and in the case of electrically neutral $\beta$-equilibrated quark matter, which is important for modelling NSs. Electric neutrality requires a proper amount of electrons with chemical potential $\mu_e = \mu_u - \mu_d$ providing $\beta$-equilibrium. At small densities where quarks are confined, we construct a phase transition of the quark matter EoS with the hadron one. We use DD2 EoS with hyperons, which agrees with the low density calculations of the chiral effective field theory from Ref. [36] and is referred to as DD2npY-T [6]. The quark and hadron EoSs are matched by means of the Maxwell construction corresponding to the first order phase transition. Such a hybrid EoS is shown in Figure 2. It is worth mentioning that the hadron-to-quark matter transition happens above the transition from chirally broken phase to the 2SC one in pure quark matter. Therefore, the quark matter branch of hybrid EoS is already color superconducting. For the considered values of $M_g$ diquark coupling strongly influences the onset density of quark matter decreasing with $\eta_D$. The stiffness of the quark matter EoS is regulated by the vector coupling. A correlated variation of $\eta_V$ and $\eta_D$ allows us to generate a family of quark-hadron EoSs consistent with the constraints obtained within the multipolytrope analysis of the observational data of PSR J1614+2230 [37] and PSR J0740+6620 [2] as well as the statistical analysis from Ref. [38].

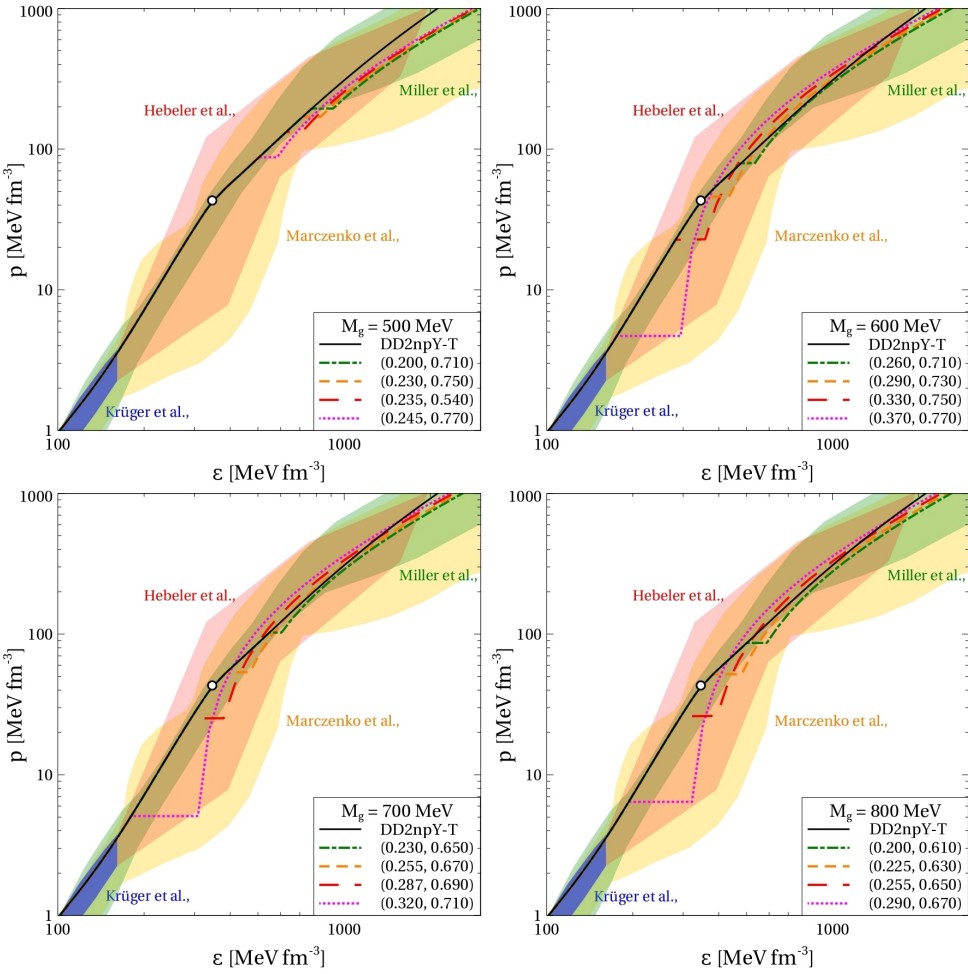

**Figure 2.** Hybrid EoS of cold electrically neutral quark-hadron matter at $\beta$-equilibrium in the plane of energy density $\varepsilon$ and pressure $p$. Empty circles on the hadronic curves indicate the hyperon onset. The shaded areas represent the nuclear matter constraints from Refs. [2,36–38] discussed in the text. The curves corresponding to hybrid EoSs are labeled with pairs of numbers $(\eta_V, \eta_D)$.

We want to point out that different values of the non-perturbative gluon mass lead to qualitatively different properties of the resulting EoS. At $M_g = 500$ MeV, the onset density

of quark matter is limited from below by about 450 MeV fm$^{-3}$ corresponding to $\eta_V = 0$ and $\eta_D = 0.77$. Smaller onset densities can be obtained only if going to the unphysical region $\eta_D > \eta_D^{max}$. At physical values of the diquark coupling, the onset of quark matter for $M_g = 500$ MeV always occurs after the hyperonization of the matter. As a result, the lower limit for the NS maximum mass can be reached only marginally, while the tidal deformability is outside the observational bounds. Already at $M_g = 600$ MeV, the onset density of quark matter is not limited from below, which provides positive feedback to the problem of fulfilling the observational constraints.

For the considered values of the vector and diquark couplings and the lower value of the gluon mass, $M_g = 500$ MeV, the quark-hadron mixed phase is located at $\varepsilon = 500$–900 MeV fm$^{-3}$. For $M_g \geq 600$ MeV, however, the mixed phase lies at lower energy densities $\varepsilon = 180 - 500$ MeV fm$^{-3}$. As shown below, small values of the non-perturbative gluon mass are inconsistent with the observational constraints on the NS mass–radius relation and the bound on the tidal deformability. Based on this, we conclude that our approach predicts the quark-hadron transition at energy densities below 500 MeV fm$^{-3}$. This range coincides with the lattice QCD results related to the chiral crossover region at vanishing chemical potential [39]. More recent analyses of the modern NS mass and radius constraints from multi-messenger observations using hybrid EoS with color superconducting quark matter do also find the hadron-to-quark matter transition at energy densities below 500 MeV fm$^{-3}$; see [40,41]. Such a coincidence of energy density domain for the phase transformation in two very different regions of the QCD phase diagram has already been observed in [42]. However, the mechanism of this transformation at finite temperatures and finite densities can substantially differ. This makes a theoretical interpretation of the above "universality" of the transition energy density challenging.

It is worth mentioning that for $M_g \geq 600$ MeV our approach is able to generate EoSs, which at high densities are significantly stiffer than the hadronic EoS DD2npY-T. At the same time, this feature is consistent with approaching the conformal limit $c_S^2 \rightarrow 1/3$. Indeed, as can be seen in Figure 3, after the hadron-to-quark matter transition, the squared speed of sound has a value of $\sim$0.5 and then decreases, approaching the conformal value at asymptotically high densities. Such a decrease of $c_S^2$ in the density range typical for NSs was recently reported based on the model agnostic statistical analysis [43]. Within our approach, the speed of sound reaches its maximal value at the quark boundary of the quark-hadron mixed phase. In other words, $c_S^2$ peaks in the color superconducting 2SC phase right after the hadron-to-quark matter transition. For the EoSs providing the best agreement with the observational constraints discussed below (red and purple curves in the upper right and lower panels in Figure 3), this maximum is located at $\varepsilon \simeq 300$–400 MeV fm$^{-3}$. Remarkably, this range of energy densities corresponding to the speed of sound maximum is in very good agreement with the results of Ref. [44] obtained within the model agnostic statistical analysis, which also respects the conformal limit. A similar peak of $c_S^2$ was also reported in Ref. [8] for the scenario of a first order phase transition FOPT-1 with the Group 1 of EoS.

For $M_g \geq 600$ MeV and energy densities interesting for the phenomenology of NSs, $c_S^2$ varies around 0.45–0.55. Such values have been obtained for color superconducting quark matter within the nonlocal Nambu–Jona–Lasinio model with covariant [45] and also with instantaneous formfactors [41]. At the same time, for a given EoS of quark matter, the relative variation of $c_S^2$ is about 10%, being in tension with the assumption of the constant speed of sound (CSS) parameterization of the quark matter EoS [46,47].

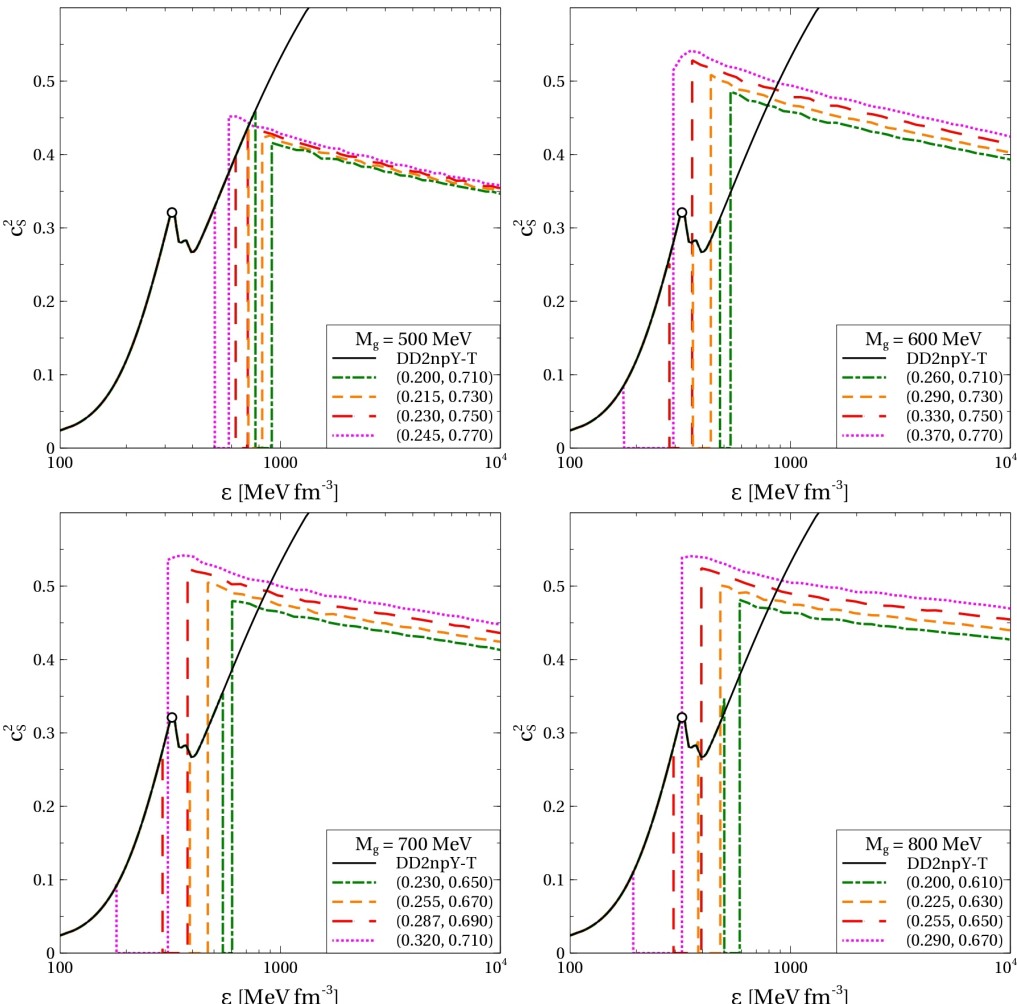

**Figure 3.** Squared speed of sound $c_S^2$ of cold electrically neutral quark-hadron matter at $\beta$-equilibrium as a function of energy density $\varepsilon$ calculated with the EoS shown in Figure 2. Empty circles on the hadronic curves indicate the hyperon onset. The curves corresponding to hybrid EoSs are labeled with pairs of numbers $(\eta_V, \eta_D)$.

## 4. Compact Stars with Quark Cores

Observational data on the masses and radii of NSs give important constraints on their EoS. These constraints include the measurement of the lower limit of the TOV maximum mass $2.01^{+0.04}_{+0.04}$ M$_\odot$ for the pulsar PSR J0348+0432 in a binary system with a white dwarf companion [48], the results of the Bayesian analysis of the observational data from PSR J0740+6620 [2,5] and PSR J0030+0451 [49,50], as well as the constraints on the masses and radii obtained from the gravitational wave signal and the kilonova light curve of the binary neutron star merger GW170817 [7,51,52]. We confronted these constraints with the mass–radius relations obtained by solving the problem of relativistic hydrostatic equilibrium, i.e., the TOV equation supplemented with the necessary boundary condition. The family of hybrid quark-hadron EoSs presented in Figure 2 was used as an input for this task. The corresponding mass–radius relations are shown in Figure 4. For $M_g = 500$ MeV, the quark matter onset mass is rather high, while the TOV maximum masses reach the observational limit, if at all, only marginally. A similar problem arises when one compares the value for $R_{1.4}$, the radius for an NS with a mass of 1.4 M$_\odot$, with the constraint $R_{1.4} = 11.75^{+0.86}_{-0.81}$ km that was derived in [1] from available multi-messenger observations. This constraint cannot be fulfilled by the models based on $M_g = 500$ MeV.

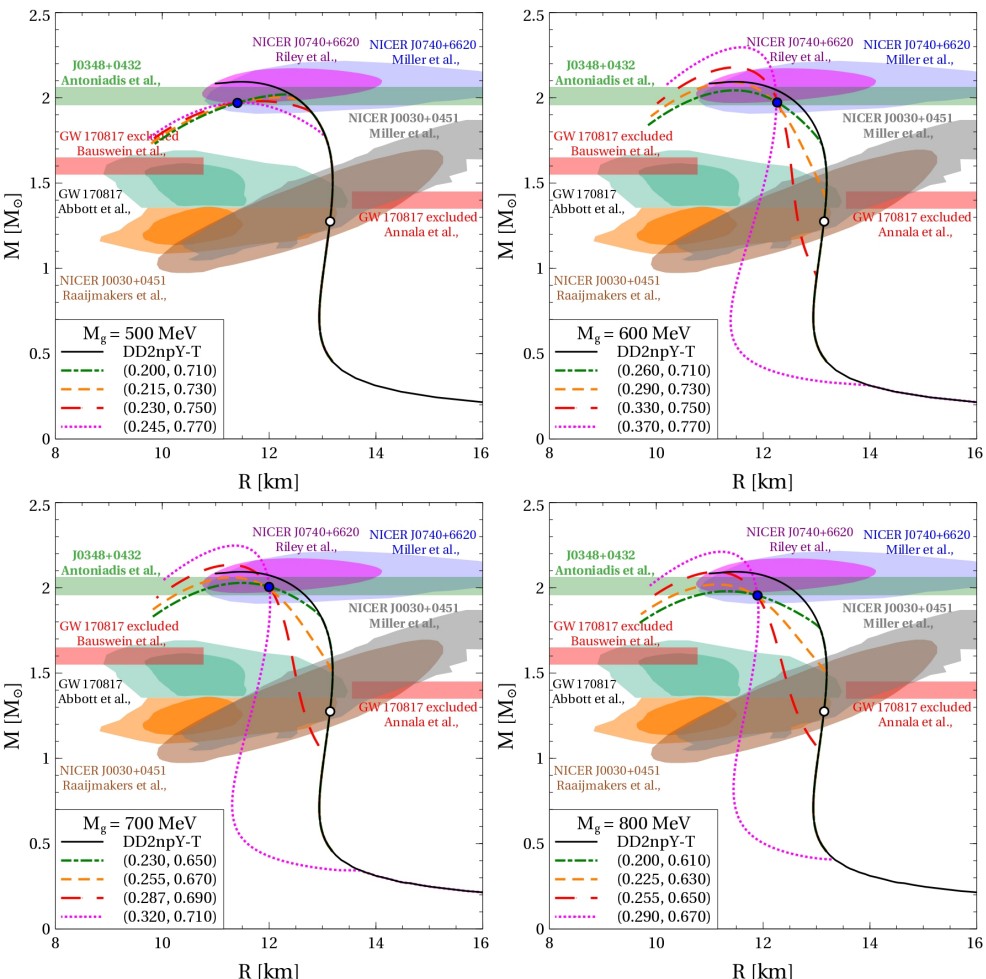

**Figure 4.** Mass–radius relation of hybrid NSs with the quark-hadron EoS presented in Figure 2. The empty circle on the hadronic curves indicates the hyperon onset. The blue filled circles represent the special points with the mass $M_{SP}$ found according to the fitting procedure described in the text. The astrophysical constraints from Refs. [2,5,7,48–52] depicted by the colored bands and shaded areas are discussed in the text. The curves corresponding to hybrid EoSs are labeled with pairs of numbers $(\eta_V, \eta_D)$.

Increasing the effective gluon mass diminishes the reduction of the vector and diquark couplings at high densities. This leads to a stiffening of the quark matter EoS and increases the maximum mass of the corresponding NS sequences. As we have shown in the previous section, increasing the gluon mass lowers the NS mass where the onset of deconfinement occurs. Therefore, these masses appear to be anticorrelated.

Lowering the onset mass of quark matter for the models with $M_g \geq 600$ MeV also provides agreement of our approach with the constraint on $R_{1.4}$. Thus, for all $M_g \geq 600$ MeV, the vector and diquark couplings can be adjusted so that all the above constraints on the mass–radius relation of NSs are respected. We also would like to stress that the best agreement with the observational data is provided for low quark matter onset masses.

A remarkable feature of the mass–radius relation of NSs with quark cores corresponds to existence of the so-called "special point" (SP), that being a narrow region of intersection of the mass–radius curves [53]. In Refs. [53–57], it was reported and thoroughly studied within the CSS parameterization of the quark-matter EoS. As can be seen from Figure 4, the SP also appears within the present approach, which cannot be phenomenologically caught by the CSS parameterization. Therefore, we conclude that the SP is likely to be a rather general feature of solutions of the TOV equation not limited to a given class of EoS of hybrid NSs.

The analysis of the gravitational wave signal from the inspiral phase of the binary neutron star merger gives direct access to the dimensionless tidal deformability $\Lambda = \frac{2}{3}k_2 C^{-5}$ expressed through the second Love number $k_2$ and the stellar compactness $C = M/R$. The measurement of the signal from the merger event GW170817 allowed to constrain the tidal deformability of a 1.4 $M_\odot$ NS to the range $\Lambda_{1.4} = 190^{+390}_{-120}$ [7]. Figure 5 shows the dependence of $\Lambda$ on M. At small values of the non-perturbative gluon mass, the agreement with the observational constraint is never achieved due to late onset of quark matter. At larger values of $M_g$, it is always possible to adjust the vector and diquark couplings in order to provide a tidal deformability in the range $\Lambda = 70$–$580$. Similar to the case of the mass–radius relation, the observational data prefer the values of the vector and diquark couplings corresponding to an early onset of quark matter.

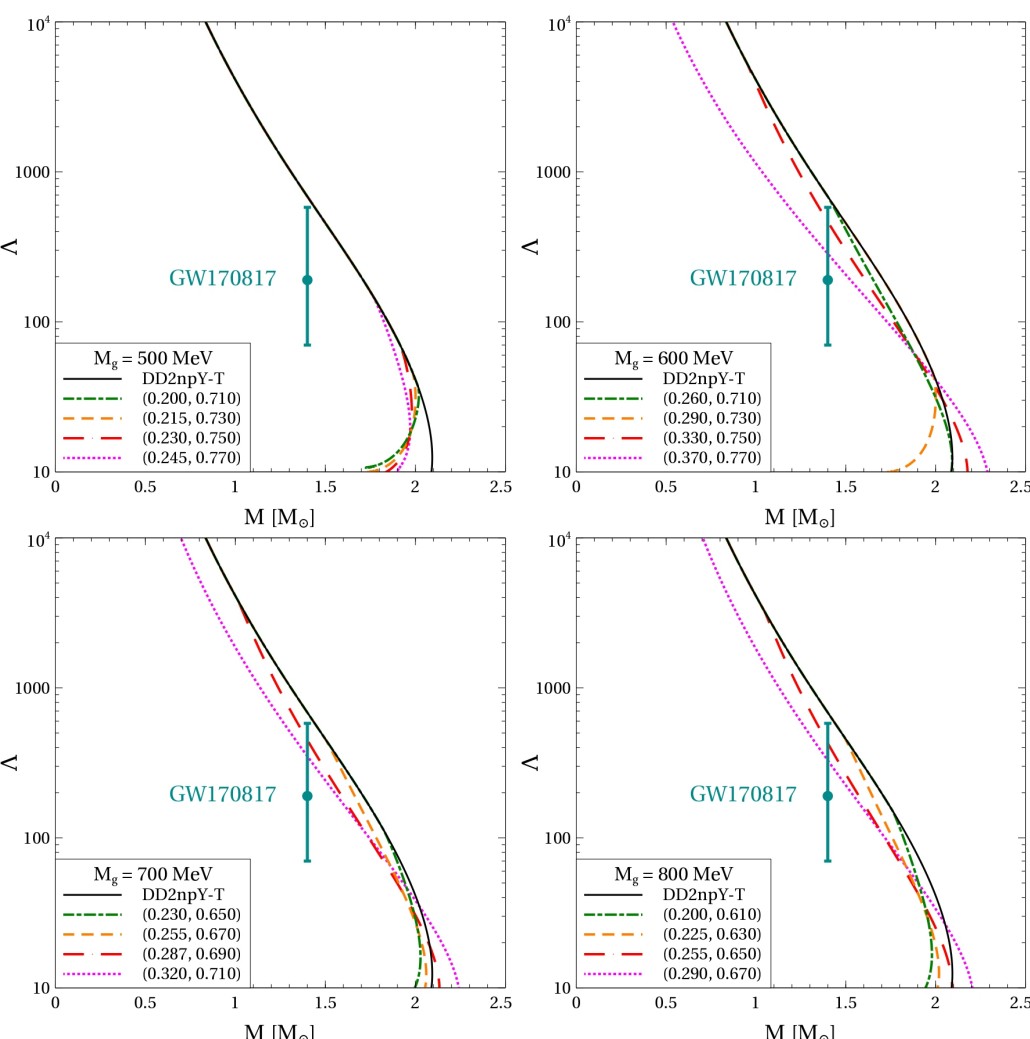

**Figure 5.** Dimensionless tidal deformability $\Lambda$ as a function of stellar mass M with the quark-hadron EoSs presented in Figure 2. The error bar corresponds to the observational constraint discussed in the text. The curves corresponding to hybrid EoSs are labeled with pairs of numbers $(\eta_V, \eta_D)$.

The possibility of conformal or near conformal behavior of matter in the cores of heavy NSs was recently discussed in Refs. [38,43]. We check this possibility within our model, which by construction respects the conformal limit at asymptotically high densities. As outlined above, the smaller the non-perturbative gluon mass, the earlier the conformality is reached. Therefore, we present an analysis for $M_g = 600$ MeV. We also consider only those parameterizations of the hybrid EoS labeled by pairs of numbers $(\eta_V, \eta_D)$, which provide consistency with the above-mentioned constraints on the NS mass–radius relation and tidal deformability. Figure 6 shows profiles of energy density and squared speed of sound for

NSs of several masses obtained for the EoSs fulfilling these selection criteria. The highest NS masses correspond to the maximum ones supported by the corresponding hybrid EoSs. Both $\varepsilon$ and $c_S^2$ exhibit a discontinuous change at the sharp interface between quark core and hadron envelope due to the strong first order phase transition. The mixed quark-hadron phase is absent since its pressure remains constant for increasing density so that there is no pressure gradient that could balance the gravitational force which compresses the matter. We note that at M = 0.6 $M_\odot$ for the EoS with $\eta_V = 0.330$ and $\eta_D = 0.750$ (left panels of Figure 6) $\varepsilon$ and $c_S^2$ are continuous since quark matter does not occur in this stellar configuration. In the case of softer EoS (left panels) the energy density reached in the center of the heaviest stars is higher compared to the case of stiffer EoS (right panels). In both cases, this central $\varepsilon$ does not exceed 1200 MeV fm$^{-3}$, which is well below the range of energy densities where quark matter approaches the conformal limit.

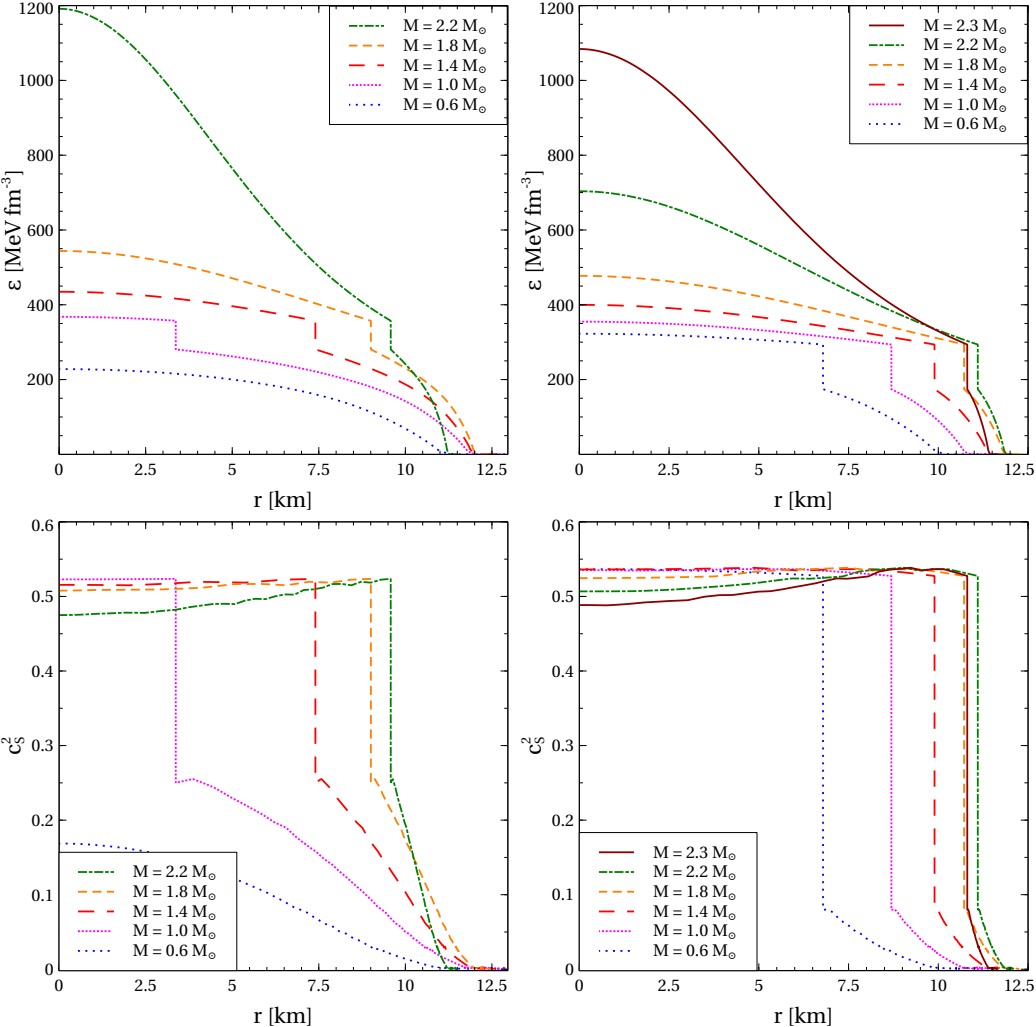

**Figure 6.** Profiles of energy density $\varepsilon$ (**upper panels**) and squared speed of sound $c_S^2$ (**lower panels**). The calculations are performed for NSs of several masses M indicated in legends, $\eta_V = 0.330$, $\eta_D = 0.750$ (**left panels**), $\eta_V = 0.370$, $\eta_D = 0.770$ (**right panels**) and $M_g = 600$ MeV. The NS masses are limited by the maximum values provided by the corresponding hybrid EoSs.

It is important to note that at higher $M_g > 600$ MeV the central value of the energy density is expected to be even smaller due to a stiffening of the quark matter EOS caused by the slower melting of the vector and diquark couplings. The behavior of $c_S^2$ supports the conclusion that quark matter inside the hybrid NSs does not reach the conformal limit. Indeed, despite decreasing towards the NS center, $c_S^2$ does not get much smaller than 0.5 even in the cores of the heaviest NSs.

Now a comment with respect to the compression modulus is in order. It quantifies the curvature of the density-dependent energy per nucleon $E/A = \varepsilon/n_B - m_N$ of nuclear matter, where $m_N$ is the nucleon mass. Thus, the compression modulus reads [58,59]

$$K_{\text{NM}} = 9n_B^2 \frac{\partial^2}{\partial n_B^2} \frac{E}{A} = 9n_B^2 \frac{\partial^2}{\partial n_B^2} \frac{\varepsilon}{n_B}. \tag{28}$$

Using the thermodynamic identity $p = n_B^2 \partial(\varepsilon/n_B)/\partial n_B$, we can replace the derivative $\partial(\varepsilon/n_B)/\partial n_B$ and arrive at $K_{\text{NM}} = 9(\partial p/\partial n_B - 2p/n_B)$. Furthermore, the density derivative of the pressure in this relation can be replaced using the relation $c_S^2 = \mu_B^{-1}\partial p/\partial n_B$, which is provided by the thermodynamic identity $\mu_B = \partial\varepsilon/\partial n_B$ and definition of $c_S^2$. Finally, utilizing $n_B = (p + \varepsilon)/\mu_B$, we obtain

$$K_{\text{NM}} = 9\mu_B \left( c_S^2 - \frac{2p}{p + \varepsilon} \right) = 9\mu_B \left( c_S^2 - \frac{2 - 6\delta}{4 - 3\delta} \right), \tag{29}$$

where in the second step the pressure was expressed as $p = (1/3 - \delta)\varepsilon$. Equation (29) relates the compression modulus to the speed of sound and interaction measure, making $K_{NM}$ a quantity useful for analyzing the possibility of reaching the conformal limit in NSs. At small densities, where quark matter has $\delta \simeq 1/3$ (see Section 3), the compression modulus attains a positive value $K_{\text{NM}} \simeq 9\mu_B c_S^2$. This signals a convex energy per baryon $E/A = \varepsilon/n_B - m_N$. At high densities, close to the conformal limit $c_S^2 \to 1/3$, $\delta \to 0$ and $K_{\text{NM}} \to -3\mu_B/2$ indicating a concave $\varepsilon/n_B \propto n_B^{1/3}$. This scaling follows from the fact that in the conformal regime $\varepsilon \propto \mu_B^4$ and $n_B \propto \mu_B^3$. Thus, $K_{\text{NM}}$ is a monotonously decreasing function of density changing from positive to negative values. Its vanishing, i.e., $K_{\text{NM}} = 0$, indicates the inflection point of $\varepsilon/n_B$, which is a precursor of the conformal regime providing $K_{\text{NM}} < 0$.

Figure 7 shows profiles of the compression modulus for several NS masses obtained for the hybrid EoSs with $M_g = 600$ MeV, which respect the observational constraints mentioned above. These profiles exhibit a discontinuous change of $K_{\text{NM}}$ due to the first order phase transition. It can be seen in Figure 2 that for the considered EoSs $p \ll \varepsilon$ at the phase transition leading to $K_{NM} \simeq 9\mu_B c_S^2$. This explains the high values of the compression modulus of quark matter in the vicinity of the deconfinement transition since the corresponding $c_S^2$ is almost an order of magnitude larger than in the case of nuclear matter at the saturation density. At small M, the energy density of NS matter remains small and the quark part of the profiles is quite flat. In the case of heavy NSs, the range of $\varepsilon$ extends to higher values and decrease of the compression modulus toward the NS center becomes prominent. This decrease is more pronounced for softer EoSs. In other words, the smaller the NS maximum mass provided by a given hybrid EoS, the smaller the values of the compression modulus reached in its center. At the same time, observational constraints on the NS mass–radius relation require quite stiff EoS. As a result, $K_{\text{NM}}$ barely vanishes, if at all, even in the centers of the heaviest NSs. For example, the heaviest stellar configuration presented on the left panel of Figure 7 yields $K_{NM} = -49$ MeV, which is negligible compared to the value of 4.9 GeV on the quark matter side of the quark deconfinement transition. Thus, within the range of densities typical for NSs $\varepsilon/n_B$ remains convex or marginally reaches the inflection point, meaning that quark matter remains far from the conformal limit.

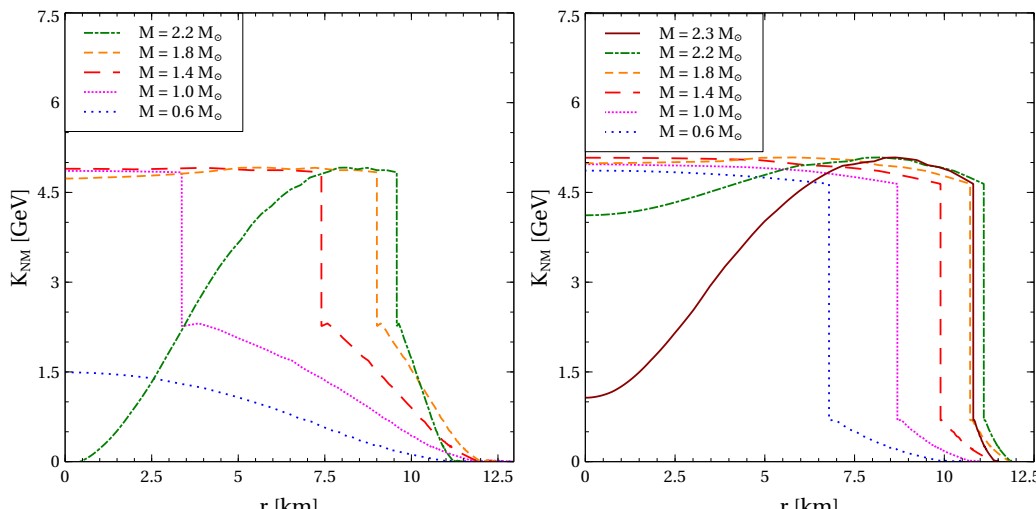

**Figure 7.** Profiles of the compression modulus $K_{NM}$ calculated for NSs of several masses M indicated in legends when $\eta_V = 0.330$, $\eta_D = 0.750$ (**left panel**) and when $\eta_V = 0.370$, $\eta_D = 0.770$ (**right panel**) for $M_g = 600$ MeV. The NS masses are limited by the maximum values provided by the corresponding hybrid EoSs.

## 5. Discussion and Conclusions

The problem of restoring the conformal limit within effective models of quark matter was given a treatment within the recently proposed relativistic density functional approach. In addition to mimicking quark confinement by a rapid growth of the quark self-energy in the confining region, the pseudo-scalar sector of this approach is equivalent to a chiral quark model with medium dependent coupling constants. In order to provide its conformal behavior at high densities, we generalized the vector repulsion and diquark pairing channels to the case when the corresponding couplings decrease with the density. We also demonstrated that a quark model with density-dependent vector and diquark couplings can be formulated within the relativistic density functional approach. The particular behavior of the vector and diquark couplings was motivated by an analysis of the quark repulsion energy due to non-perturbative gluon exchange in QCD in the Landau gauge. We showed that the conformal limit is asymptotically reached within the present approach at any finite value of the non-perturbative gluon mass.

The developed approach was applied in order to Maxwell-construct a family of hybrid quark-hadron EoSs used to model NS. We found that observational data prefer the non-perturbative gluon mass exceeding a value above 500 MeV. Another general conclusion of our analysis is that color superconductivity lowers the onset density of quark matter and that such early quark deconfinement is favoured by the observational constraints on the mass–radius relation and tidal deformability of NSs. More precisely, our analysis supports the hadron-to-quark matter transition at energy densities within the range 180–500 MeV fm$^{-3}$. We also report that energy density reached in the cores of the heaviest NSs is far from the region of conformality of quark matter.

**Author Contributions:** Conceptualization, methodology, investigation, writing—original draft preparation and editing, O.I. and D.B.B.; software, O.I.; funding acquisition, D.B.B. All authors have read and agreed to the published version of the manuscript.

**Funding:** This work was supported by the Polish National Science Centre (NCN) under grant No. 2019/33/B/ST9/03059. The work was performed within a project that has received funding from the European Union's Horizon 2020 research and innovation program under grant agreement STRONG–2020—No 824093.

**Informed Consent Statement:** Not applicable.

**Data Availability Statement:** Not applicable.

**Acknowledgments:** We acknowledge discussions with Larry McLerran, Michal Marczenko and Krzysztof Redlich.

**Conflicts of Interest:** The authors declare no conflict of interest.

## Appendix A

In order to analyze the minimum of the thermodynamic potential with respect to chiral condensate, vector field and diquark pairing gap, we notice that $\Omega$ given by Equation (12) explicitly depends on seven variables, i.e., $\mu_u$, $\mu_d$, $\langle q^+ q \rangle$, $\langle \bar{q} q \rangle$, $|\langle \bar{q}^c i \tau_2 \gamma_5 \lambda_2 q \rangle|$, $\omega$ and $\Delta$. The part of quark quasiparticles is a function of all these variables except the quark number density and diquark condensate. Therefore, its full differential can be written as

$$d\Omega_q = \sum_f \frac{\partial \Omega_q}{\partial \mu_f}(d\mu_f + d\omega) + \frac{\partial \Omega_q}{\partial m}\frac{\partial \Sigma_{MF}}{\partial \langle \bar{q} q \rangle}d\langle \bar{q} q \rangle + \frac{\partial \Omega_q}{\partial \Delta}d\Delta. \tag{A1}$$

Here, we accounted for the fact that the vector field and the chiral condensate enter $\Omega_q$ through the effective chemical potential $\mu_f^* = \mu_f + \omega$ and the effective mass $m^* = m + \Sigma_{MF}$, respectively. The second and the third terms in Equation (12) depend only on $\langle \bar{q} q \rangle$ yielding

$$d(\mathcal{U}_{MF} - \langle \bar{q} q \rangle \Sigma_{MF}) = -\langle \bar{q} q \rangle \frac{\partial \Sigma_{MF}}{\partial \langle \bar{q} q \rangle}d\langle \bar{q} q \rangle, \tag{A2}$$

where we used definition of the quark mean-field self-energy given by Equation (5). The fourth term in the expression for the thermodynamic potential (12) is a function of $\omega$ and $\langle q^+ q \rangle$, entering it through the vector coupling. Thus

$$d\left(-\frac{\omega^2}{4G_V}\right) = -\frac{\omega}{2G_V}d\omega + \frac{\omega^2}{4G_V^2}\frac{\partial G_V}{\partial \langle q^+ q \rangle}d\langle q^+ q \rangle. \tag{A3}$$

Similarly, for the fifth term we obtain

$$d\left(\frac{\Delta^2}{4G_D}\right) = \frac{\Delta}{2G_D}d\Delta - \frac{\Delta^2}{4G_D^2}\frac{\partial G_D}{\partial |\langle \bar{q}^c i \tau_2 \gamma_5 \lambda_2 q \rangle|}d|\langle \bar{q}^c i \tau_2 \gamma_5 \lambda_2 q \rangle|. \tag{A4}$$

Finally, differentials of the rearrangement terms $\Theta_V$ and $\Theta_D$ are directly found from Equation (17)

$$d\Theta_V = \langle q^+ q \rangle^2 \frac{\partial G_V}{\partial \langle q^+ q \rangle}d\langle q^+ q \rangle, \tag{A5}$$

$$d\Theta_D = |\langle \bar{q}^c i \tau_2 \gamma_5 \lambda_2 q \rangle|^2 \frac{\partial G_D}{\partial |\langle \bar{q}^c i \tau_2 \gamma_5 \lambda_2 q \rangle|}d|\langle \bar{q}^c i \tau_2 \gamma_5 \lambda_2 q \rangle|. \tag{A6}$$

The next step corresponds to considering the total differential of the thermodynamic potential at constant quark chemical potentials ($d\mu_u = d\mu_d = 0$). We note that in this case the differentials $d\langle q^+ q \rangle$ and $d|\langle \bar{q}^c i \tau_2 \gamma_5 \lambda_2 q \rangle|$ do not necessarily vanish due to variation of

chiral condensate, vector field and diquark pairing gap. Thus, using Equations (A1)–(A6) differential of the thermodynamic potential can be written as

$$
\begin{aligned}
d\Omega \;=\; & \left( \sum_f \frac{\partial \Omega_q}{\partial \mu_f} - \frac{\omega}{2G_V} \right) d\omega \\
& + \left( \frac{\omega^2}{4G_V^2} - \langle q^+ q \rangle^2 \right) \frac{\partial G_V}{\partial \langle q^+ q \rangle} d\langle q^+ q \rangle \\
& + \left( \frac{\partial \Omega_q}{\partial m} - \langle \bar{q} q \rangle \right) \frac{\partial \Sigma_{MF}}{\partial \langle \bar{q} q \rangle} d\langle \bar{q} q \rangle \\
& + \left( \frac{\partial \Omega_q}{\partial \Delta} + \frac{\Delta}{2G_D} \right) d\Delta \\
& - \left( \frac{\Delta^2}{4G_D^2} - |\langle \bar{q}^c i\tau_2 \gamma_5 \lambda_2 q \rangle|^2 \right) \frac{\partial G_D}{\partial |\langle \bar{q}^c i\tau_2 \gamma_5 \lambda_2 q \rangle|} d|\langle \bar{q}^c i\tau_2 \gamma_5 \lambda_2 q \rangle|.
\end{aligned}
\tag{A7}
$$

The conditions to minimize the thermodynamic potential can be found by requiring zero values of the coefficients near the differentials of the above mentioned variables of $\Omega$. The second line of Equation (A7) yields the mean-field equation for the vector field, i.e., $\omega = -2G_V \langle q^+ q \rangle$. With this result and $\langle q^+ q \rangle = \sum_f \langle f^+ f \rangle$, we conclude from the first line of Equation (A7) that $\langle f^+ f \rangle = -\partial \Omega_q / \partial \mu_f$. Direct differentiation of Equation (13) with respect to $\mu_f$ yields the expression for number density of a given quark flavor (18). It is seen from the third line of Equation (A7) that the mean-field equation for chiral condensate is $\langle \bar{q} q \rangle = \partial \Omega_q / \partial m$. One arrives at Equation (19) by directly calculating the partial derivative of $\Omega_q$ with respect to the current quark mass. From the fourth and fifth lines of the expression for $d\Omega$ we immediately recover the pairing gap equation $\Delta = 2G_D |\langle \bar{q}^c i\tau_2 \gamma_5 \lambda_2 q \rangle|$ and the diquark condensate $|\langle \bar{q}^c i\tau_2 \gamma_5 \lambda_2 q \rangle| = -\partial \Omega_q / \partial \Delta$. The latter can be given the form of Equation (20) by finding the partial derivative of the thermodynamic potential of quark quasiparticles with respect to the diquark paining gap.

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
