# Peer review of "Recovering the Conformal Limit of Color Superconducting Quark Matter within a Confining Density Functional Approach"

_2571-712X, doi:10.3390/particles5040038_

Round 1

Reviewer 1 Report

Fascinating paper. Well written and of interest to the particle physics and astrophysics communities. The extension of the DFT approach to density dependent vector and diquark couplings is important. Analogous to what practitioners of DFT do for electronic structure calculations, the work is able to investigate  the equation of state of quark matter relevant to neutron stars. as a function of gluon mass. Comparison of their results to observational data yields a higher value of the gluon mass which particle physicists will find interesting. How consistent are the gluon mass constraints provided by neutron stars consistent with other non-astronomical constraints provided in particle physics. Particularly the older paper by Graziani (1987, Z. Phys. C - Particles and Fields 33) where he applied massive gauge invariant QCD to the Shifman, Vainshtein, and Zakharov expansion of the two-point current correlation functions to obtain a gluon mass.  Are there similar calculations and perhaps particle physics experiments that also provide a limit to the gluon mass and are they consistent with what the the astronomical constraints provide?

Reviewer 2 Report

This manuscript reports on a an effective mean-field NJL-like model for quark matter with density dependent vector and di-quark couplings, where the density dependence is constrained such as to recover the conformal limit at high density, with three residual free parameters linked to the vacuum value of the two couplings, and the non-perturbative gluon mass. The model is coupled to the hadronic DD2 EoS with hyperons, to build hybrid star equations of state. The corresponding M(R) curves obtained by varying the 3 free parameters are compared to present astrophysical    constraints, and confirm previous findings with similar modelings, namely the occurrence of an early phase transition in an energy density range between 180 and 500 MeV/fm3.

The paper is well written, the results are interesting,  and only minor modifications are needed in my opinion before publication:

1) The sentence in the introduction 

"These new constraints on the NS mass-radius relation pose a challenge for purely hadronic EoS which now are not quite excluded, but become marginal. In particular because of the appearance of hyperons and heavier (multi-)baryon states at NS masses above ∼ 1.4 M⊙, because of their effect to soften the EoS and thus to lower the maximum mass and the radii of NS which leads to the "hyperon puzzle"."  should be suppressed or deeply revised. Indeeed, purely hadronic equations of state have been shown by many authors to perfectly satisfy the present astrophysical constraints, see for instance P.Pang et al ApJ922,1,14;  R.Somasundaram et al,arXiv:2112.08157; H.Dinh Thi et al, Universe,7,10,373. Concerning the "hyperon puzzle", it is nowadays well recognized that the couplings to the hyperons do not necessarily have to respect simplistic SU(6) symmetry considerations, as shown by the very fact that the hadronic EoS chosen by the authors for this application does include hyperonic degrees of freedom and at the same time respects the 2Mo constraint.

2) the meaning of the different lines in figs.2 and 3 is not clear. I guess the numbers refer to the valuesof eta_V and eta_D? This should be specified in the captions

 3) the fact that the value of the transition energy density found by the authors is close to the one found by LQCD calculation at zero chemical potential is observed by the authors as a sign of "universality" of the phase transition. Why should one expect that the QCD finite temperature effect should act in the same way as for finite density? I would rather think that it is just a coincidence. If not, an explanation should be given. 

Reviewer 3 Report

The authors of “RECOVERING THE CONFORMAL LIMIT OF COLOR SUPERCONDUCTING QUARK MATTER WITHIN A CONFINING DENSITY FUNCTIONAL APPROACH” develop a strategy to reproduce the conformal limit of QCD at high energy scales. In doing this, they use a NJL-like model with G_D and G_V coupling constant depending on the scalar and diqaurk condensates.

Although the proposed work is interesting, I have some doubts about the used approach.

1) In the framework of the density functional approach I would have expected that
the grand potential in Eq.(12) would be a functional of  <q^\dagger q>, <\bar q q> etc.

In the present approach this should imply, for instance, that both \omega and G_V are functional of  <q^\dagger q> and that the minimization should be done with respect to the function <q^\dagger q>. A similar argument holds for \Delta.  This does not lead to the gap equations (18), (19) and (20). In other words, it seems that in deriving these equations the authors did not include the dependence  of G_V and G_D on the corresponding condensates as reported in (23) and (24).

2) It would be better to have a speed of sound and a compressibility profile for a few NS configurations. The reason is that it is not really clear which part of the plots reported in Figs. 1 and 3 is relevant for compact stars and whether the compressibility has the right behavior inside the star.

3) It is not clear to me how to reconcile the behavior reported in the two panels of Fig.1

i) Given the definition of the interaction measure, it seems  that

P = (1/3 -  \delta) \epsilon,

Thus, when \delta = 1/3 one correctly has that c_s^2 =0. This is fine.
But then,  when c_s^2 > 1/3 it seems that \delta should be negative. Since the largest value of c_s is reached at the transition point, around \mu_B = 1.1 GeV ,  it seems that at  that point \delta should be negative.

ii) Around \mu_B = 1.1 GeV the interaction measure vanishes. Since the interaction measure is basically the trace of the energy momentum tensor, it means that the system is conformal. This does not sound correct.
